# Beyond the counter: Pharmacists' preparedness and response strategies in terrorism-related emergencies in Quetta, Pakistan

Fahad Saleem[1,2], Fazal ur Rehman Khilji[1], Sajjad Haider[1], Qaiser Iqbal[1], Baharudin Ibrahim[2], Fatiha Hana Shabaruddin[2], Mohammad Bashaar [3]*

1 Faculty of Pharmacy & Health Sciences, University of Balochistan, Quetta, Pakistan, 2 Faculty of Pharmacy, Universiti Malaya, Kuala Lumpur, Malaysia, 3 SMART Afghan International Trainings and Consultancy, Kabul, Afghanistan

* dr.mbashaar@gmail.com

## Abstract

Terrorism-related disasters (TRDs) continue to exert profound and recurring pressures on healthcare systems, particularly in vulnerable regions like Pakistan. Although pharmacists are increasingly recognized as an essential component of disaster management, there is a clear gap in the literature regarding their preparedness, experience, and specific roles in responding to TRDs particularly in low and middle-income countries. This study aimed to explore the preparedness, experiences, and response strategies of pharmacists managing TRDs at the Trauma Centre of Sandeman Provincial Hospital, Quetta, Pakistan. A qualitative design was adopted, guided by the Consolidated Criteria for Reporting Qualitative Research. Semi-structured, face-to-face interviews were conducted with pharmacists (n = 10) providing services at the Trauma Centre. Data were audio-recorded, transcribed verbatim, validated by participants, and analyzed using thematic content analysis. Analysis revealed five overarching themes: (1) pharmacists' experiences with terrorism-related incidents and existing response mechanisms; (2) professional and personal responses to emergencies, reflecting both commitment and psychological burden; (3) preparedness challenges, including lack of disaster management training, limited awareness of policies and protocols, and inadequate understanding of triage and coordination; (4) barriers such as security risks, pharmacy curriculum deficiencies, insufficient experiential learning, and minimal involvement in planning and management activities; and (5) recommendations for strengthening capacity, including revising curricula, implementing structured training programs, conducting regular disaster drills, and expanding pharmacists' roles in preparedness and response. Findings revealed a pronounced lack of formal training in disaster management, limited awareness of protocols and triage systems, and minimal involvement of pharmacists in planning and coordination activities. Despite strong professional commitment

**Data availability statement:** The data supporting the findings of this study are not publicly available due to ethical restrictions and the sensitive nature of the information collected, which could compromise participant confidentiality. Data access may be granted upon reasonable request and subject to approval by the Graduate Studies Office, University of Balochistan, which serves as the independent institutional data access authority. Contact for data access requests: Graduate Studies Office, University of Balochistan, Quetta; Email: gso@um.uob.edu.pk The dataset was collected as part of the study "Pharmacists' Preparedness and Response Strategies in Terrorism-Related Emergencies in Quetta, Pakistan", conducted at the University of Balochistan, under ethics approval IRB/FoP&HS/55/24 (dated 1 November 2024). The data are securely stored at the Graduate Studies Office, University of Balochistan, Quetta, in accordance with institutional ethical and data governance policies.

**Funding:** The author(s) received no specific funding for this work.

**Competing interests:** The authors have declared that no competing interests exist.

and frontline engagement, pharmacists' contribution remain constrained by educational, structural, and policy-level shortcomings. The study highlights the urgent need for integrating disaster management into pharmacy curricula, implementing structured training programs and regular disaster drills, and expanding pharmacists' roles within institutional and national disaster preparedness frameworks.

## Introduction

Defined as a serious disruption of communal functions and activities that results in irretrievable losses [1], disasters are recognized as a universal calamity. Throughout history, humanity has faced both natural and man-made disasters; however, their increasing frequency over the past fifty years has become a global concern [2,3]. Complementing this, the World Disasters Report highlighted a continual rise in disaster frequency in recent years, leading to significant economic losses and adverse impacts on human life and society [4].

Within this context, terrorism-related disasters (TRDs) have the unique characteristic of causing deliberate harm to society [5]. According to the Global Terrorism Database, the use of violence to attain specific goals through creating an environment of fear and coercion is termed as TRDs [6]. Terrorism-related disasters create an atmosphere of fear and result in significant physical and psychological harm. Unfortunately, the frequency of TRDs is on the rise, and their occurrence in affected countries has increased at an alarming rate [7]. In 2024, Burkina Faso was the most affected by TRDs, followed by Pakistan and Syria, with Pakistan exhibiting the largest year-on-year increase in death tolls compared to 2023 [7].

Terrorism-related disasters adversely affect the economy, social fabric, and political stability. In parallel, they cause persistent damage to infrastructure and lead to profound social consequences, including loss of life, displacement, and psychological trauma [8,9]. Where TRDs cause calamitous destruction, undeniably, the most affected domain of society is the healthcare system. Terrorism-related disasters have both immediate and long-term consequences on the healthcare systems, as they overwhelm emergency services and disrupt access to healthcare. Furthermore, the recurrent occurrence of TRDs exacerbates existing health challenges, making the commotion and interruption of healthcare services inevitable [10,11]. Therefore, the effective management of TRDs is imperative, requiring both optimal preparedness and rapid response to terrorism-related events. Such measures are crucial to curtail loss of life, minimize economic damage, and foster community resilience. Consequently, healthcare systems and professionals must be adequately trained and equipped with the necessary skills in advance to manage TRDs; otherwise, the consequences may be severely detrimental.

Focusing on the preparedness and response of healthcare professionals during TRDs, the role of pharmacists is becoming increasingly significant. The American Society of Health-System Pharmacists recommends that in the event of a terrorist attack, pharmacists must collaborate with other healthcare professionals to ensure

effective drug therapy management for victims [12]. Furthermore, the International Pharmaceutical Federation empha-sizes that crisis planning groups must include pharmacists when developing action plans and protocols for terrorism-related disaster management [13]. The International Pharmaceutical Federation also insisted on involving pharmacists in preparedness, response, and recovery phases during or after TRDs. However, to date, pharmacists' involvement and performance in terrorism-related disaster management remain largely undocumented in the literature. Supporting this observation, McCourt and colleagues, in their systematic review, reported a lack of evidence regarding pharmacists' pre-paredness for disaster management. They further noted that pharmacists' readiness to manage disasters appears to be low, and that inadequately prepared pharmacists may negatively affect patient outcomes and the overall disaster manage-ment process [14]. Likewise, gaps in pharmacists' preparedness for emergency events and disasters were also identified by a study from the United States [15]. Based on the evidence, it is crucial to ensure that pharmacists are prepared to respond to a disaster, including TRDs.

While TRDs necessitate a coordinated, multidisciplinary healthcare response, pharmacists contribute a distinct and complementary set of competencies that extend beyond those of other healthcare professionals [12,13]. Unlike physicians and nurses, whose primary responsibilities centre on clinical assessment, triage, and direct patient care, pharmacists play a pivotal role in ensuring the continuity, safety, and optimization of pharmacotherapy during mass-casualty incidents [16]. This includes rapid procurement and distribution of essential medicines, management of antidotes and emergency drug kits, prevention of medication errors under high-pressure conditions, and real-time support for rational drug selection amid resource constraints [17]. Despite these unique contributions, pharmacists are often underutilized and insufficiently inte-grated into disaster preparedness frameworks, particularly in low- and middle-income countries such as Pakistan. Exist-ing research has primarily focused on pharmacists' involvement in natural disasters and public health emergencies, with insufficient attention to the unique operational and security challenges posed by terrorism-related mass-casualty incidents [18–20]. However, pharmacists remain underutilized in disaster preparedness frameworks, particularly in conflict-affected settings. This gap is especially concerning in Pakistan, which has experienced a marked escalation in the frequency and lethality of TRDs.

Pharmacists' preparedness and proficiency in managing mass-casualty events involve training, activities, programs, and systems developed and implemented prior to the occurrence of such events. The primary objective is to anticipate unfortunate events, deliver essential medical care to victims, and mitigate both short- and long-term adverse impacts. These programs and exercises can be particularly effective in regions such as Pakistan, which faces recurrent waves of terrorism. Consequently, assessing the preparedness and response of healthcare professionals is imperative to minimize damage during TRDs in such high-risk areas. Within this context, we have previously undertaken studies, published else-where, that examined the preparedness and response of nurses and physicians toward TRDM in Pakistan [10,11]. How-ever, no systematic study has yet examined the preparedness and readiness of pharmacists in managing mass-casualty incidents resulting from terrorism. Consequently, little is known about the knowledge, skills, and professional competen-cies of pharmacists involved in terrorism-related disaster management in Pakistan. Accordingly, this study investigates the preparedness, experience, and specific response strategies of pharmacists in managing terrorism-related emergencies at the Trauma Centre of Sandeman Provincial Hospital, Quetta, Pakistan.

## Methods

### Study design

This study was designed and reported in accordance with the Consolidated Criteria for Reporting Qualitative Research (COREQ) checklist, which provides a 32-item framework to enhance transparency in the conduct and reporting of interview-based studies [21]. A completed COREQ checklist indicating where each item is addressed in the manuscript is provided in the supplementary material.

 

A qualitative approach was adopted, and face-to-face interviews were conducted. The reasons for adopting a qualitative design were clear. Qualitative methods allow researchers to explore and understand complex phenomena when prior evidence is limited [22,23]. Furthermore, these methods provide an in-depth understanding of the phenomenon under study and aid in generating hypotheses where little is known beforehand. As assessing pharmacists' readiness and response towards terrorism-related disaster management is an under-discovered research area, we believed that qualitative methods were the most suitable choice for inductive research approaches compared with other models, which is also supported by the literature [24,25]. Given the complexity of the phenomenon and the scarcity of existing evidence, a qualitative approach was appropriate to capture the nuanced experiences, perceptions, and response practices of pharmacists managing TRDs.

## Study settings

Pharmacists stationed and providing services at the Trauma Centre (TC) of the Sandeman Provincial Hospital (SPH) Quetta, were approached for the interviews. Established in 2016, the TC is an advanced and well-equipped institute that offers emergency healthcare services 24/7 with a special focus on TRDs. The TC is the only designated centre in Balochistan province with a dedicated team of healthcare professionals who are experts in treating critical injuries and major trauma cases, including victims of terrorism [10].

## Study participants, criteria, and sampling

Pharmacists registered with the Pharmacy Council of Balochistan, stationed and providing services at the TC and consenting to participate were included in the study. Since our primary focus was on pharmacists, we adopted a universal sampling approach, and all pharmacists (n = 14) of the TC were approached by the research team [26]. Universal sampling is recommended in qualitative research when the population of interest is limited and possesses unique experiential knowledge that is not readily accessible elsewhere [24]. Given the specialized nature of terrorism-related disaster response at the TC, inclusion of all eligible pharmacists was considered methodologically appropriate to ensure maximum depth and contextual richness of data. Pharmacists on rotations, on daily wages/contract and not willing to participate were excluded because including such participants could have introduced experiential heterogeneity, potentially diluting the depth of insights related to institutional preparedness, coordination mechanisms, and role continuity during repeated terrorism-related incidents [23,24,27].

## The interview guide (development, validation, and piloting)

A semi-structured interview guide was conducted after an extensive literature review [10,11,28–30]. Parallel to the literature review, we also organised expert panel discussion sessions and experience sharing of healthcare professionals with experiences of managing TRDs [31,32]. The interview guide was purposely generated with open-ended questions that were meant to encourage comprehensive and nuanced responses from the pharmacists. Pharmacists were also given freedom in providing their own narratives and sharing experiences related to terrorism-related disaster management.

The guide was constructed in the English language and was subjected to face and content validity through an expert review session comprising six senior pharmacists. The expert panel showed few reservations regarding the content of the interview guide. The research team addressed the reservations, and the guide was presented for a second review session. Once approved through mutual consensus by the expert panel, the guide was piloted with three pharmacists. Analysis of the pilot scripts and data reported that the interview guide was appropriate, and participants had no issues in answering or comprehending the questions. The guide was finalized during a discussion session with the research team and was subsequently made available for the study. We did not include the pilot data in the final analysis.

### Field study (interview procedure, data collection, and analysis)

The pilot phase was conducted over a two-week period from 01/01/2025 to 15/01/2025. Participant recruitment for the field study commenced on 25/01/2025. Interviews continued until data saturation was achieved, and recruitment concluded on 10/04/2025.

The second author (FUR; PhD; Hospital Pharmacist) conducted the interviews at a designated space of the TC. The third author (SH; PhD; Lecturer) served as an observer, while the fourth author (QI; PhD; Lecturer) assisted in monitoring field notes, facial expressions, and body language to complement the interviews. Following an ice-breaking session, the interviewees were briefed on the research objectives. The interviews, each lasting approximately 30 minutes, were audio-recorded and conducted until thematic saturation was reached [22]. A debriefing session was conducted with the participants before closing the sessions. All interviews and sessions were conducted in English language.

A phenomenological framework guided the analysis, and a thematic content analysis was conducted to identify, organize, and interpret recurring patterns and meanings within the data. The recordings were transcribed verbatim by the research team. The scripts were presented to the interviewees for confirmation and approval. Following approval, the scripts underwent thematic content analysis in accordance with a standard reference framework [22]. NVivo® was used for iterative coding and analysis with any inconsistencies resolved through mutual consensus [33]. The software organized and managed the interview transcripts and supported systematic, line-by-line inductive coding. Initial codes were generated and grouped into nodes, which were iteratively compared and refined. Related nodes were clustered to develop themes and subthemes, with coding and theme development reviewed and finalized through team discussion and consensus, enhancing the rigor and transparency of the analysis [33].

Four data coders coded the data. Interviews were coded line-by-line, and an initial list of nodes was developed. Later, this was augmented in developing the framework, and transcripts were coded accordingly. New emerging nodes were added to the existing list and were categorized as emerging themes. Emerging themes and subthemes were thoroughly discussed among the research team to ensure accuracy and were subsequently used to guide data analysis and interpretation.

### Ethical approval

The study protocol was approved by the Institutional Review Board at the Faculty of Pharmacy & Health Sciences, University of Balochistan (IRB/FoP&HS/55/24). Written informed consent for participation and publication was obtained from all participants prior to the interviews. Participants were briefed on the nature of the research, assured of the confidentiality of their responses, and informed of their right to withdraw from the study at any time.

## Results

### Demographic characteristics of the study respondents

Fourteen pharmacists were approached, of whom ten participated in the interviews, including eight (80%) males. Six respondents had less than 10 years of overall experience as hospital pharmacists. Specifically, regarding their work at the TC, seven pharmacists (70%) had more than 18 months of experience. Notably, none of the participants had received formal training in DM, including TRDM, as summarized in Table 1.

The saturation was achieved at the eighth interview, but two additional interviews were conducted to ensure the saturation [34,35]. Thematic content analysis resulted in five major and nine subthemes as shown in Table 2.

**Theme 1: Experience with Terrorism-Related Incidents and Response Protocols.** This theme explores pharmacists' experiences with terrorism-related disasters and their corresponding responses. All pharmacists revealed that they have experienced, responded to, and managed different types of TRDs at TC. Among those, improvised explosive devices and suicidal attacks were repeatedly mentioned by most of the pharmacists. The suicide incidents of August 2016 and November 2024 were specifically mentioned as having collectively resulted in >100 deaths

**Table 1. Demographic characteristics of the respondents.**

| Characteristics | Frequency | Percentage |
|---|---|---|
| Age (years) | | |
| 30-40 | 7 | 70 |
| 41-50 | 3 | 30 |
| Gender | | |
| Male | 8 | 80 |
| Female | 2 | 20 |
| Education | | |
| B. Pharm | 3 | 30 |
| Pharm-D | 6 | 60 |
| M. Phil | 1 | 10 |
| Overall experience (years) | | |
| 1-5 | 6 | 60 |
| 6-10 | 2 | 20 |
| > 10 | 2 | 20 |
| Experience at Trauma centre (months) | | |
| 6-12 | 1 | 10 |
| 13-18 | 2 | 20 |
| > 18 | 7 | 70 |
| Training/ courses in Disaster Management None | 10 | 100 |

**Table 2. Themes and subthemes identified during data analysis.**

| Themes | Subthemes |
|---|---|
| **Theme 1**: Experience with Terrorism-Related Incidents and Response Protocols | |
| **Theme 2**: Pharmacists' Responses to Terrorism-Related Events | 2(a): Professional response |
| | 2(b): Personal response |
| **Theme 3**: Preparing for terrorism-related events | 3(a): Knowledge and understanding of terrorism-related disaster management protocol and policies |
| | 3(b): The gaps in managing terrorism-related disasters |
| | 3(c): Triage, assessment and coordination |
| **Theme 4**: Barriers and limitations | 4(a): Security issues (internal and external) |
| | 4(b): Curriculum deficiencies |
| | 4(c): Inadequate experiential learning opportunities |
| | 4(d): Minimal pharmacist contribution to planning and management activities |
| **Theme 5**: Suggestions and recommendations | |

and countless injuries. In addition to the damaged infrastructure, the fear and panic in the city were sickening and unimaginable.

> *"The 2024 event is something I can never forget. It was terrifying and horrifying for all of us at the TC. I was very close to the bomb blast, just a short distance away in a crowded part of the city. The sound was so overwhelming that it felt as if the explosion had happened right at the TC. Many lives were lost that day."* (Pharmacist 2, Male)

Another pharmacist (male) shared his experience that he was at the TC when the Bethel Memorial Methodist Church was attacked. *"The blast, and then the mass shooting, created such a horrible and overwhelming atmosphere - not just for the victims but for all of us at the TC. It was truly an awful and disturbing experience for the whole city (Pharmacist 9, Male)."*

It is crucial that, when a terrorism-related disaster occurs, the healthcare team is informed immediately and without delay. Timely communication ensures a quick, efficient, and effective response to the event. To achieve this, healthcare centres, allied institutions, and related organizations must adopt advanced technological information systems. In parallel, local governments should take proactive measures to strengthen information sharing during such calamities. The pharmacists were asked about the source of information and the call-up mechanism in their institute during TRDs.

> *"For situations like this, we rely on a dedicated WhatsApp group. As soon as a disaster happens, the Department of Health posts an alert there, with the directions we need to take immediate action."* (Pharmacist 4, Male)

During the interviews, we asked the respondents about the call-up mechanism contingent upon terrorism-related disaster emergencies. Pharmacists of the current study were satisfied with the response mechanism in contacting the healthcare professionals during an emergent condition.

> *"At the TC, we work around the clock on rotation. Whoever is on duty responds right away when a disaster happens. As part of our job, we report to the TC as soon as we get the information, and the duty pharmacist also calls the rest of us to make sure we're on our way. Usually, all of us manage to reach the TC within minutes."* (Pharmacist 8, Male)

**Theme 2: Pharmacists' Responses to Terrorism-Related Disasters.** While dealing with TRDs, an efficient and resourceful response of the healthcare system saves lives and curtails health risks. A rapid response results in timely care of the critically affected individuals, prevents debilities, and limits psychological traumas. In short, a well-coordinated response helps in utilizing scarce resources, supports calming the distressing environment, and builds societal confidence [36]. In the case of TRDs, a quick and effective response ensures that victims receive necessary care and have equitable access to services, minimizing further harm and promoting recovery [37]. Subsequently, this theme examines pharmacists' responses to TRDs while delivering services at the Trauma Centre.

Pharmacists of the current study reported feeling prepared while responding to TRDs both professionally and psychologically. However, certain deficiencies and reservations were also observed during the interviews. The responses are discussed as subthemes accordingly.

*Subtheme 2(a): Professional response*

Respondents of our study agreed on the importance of a professional response in disaster management. They believed a professional response is critical for ensuring that victims receive the appropriate care in a timely and efficient manner, which can significantly reduce both immediate and long-term harm. Accordingly, a pharmacist responded that *"the moment we receive an emergency alert, medicine trolleys filled with essential medicines, surgical items, and disposables are placed at the ER, OTs, and wards. The chief pharmacist supervises this process to ensure everything is ready before patients come in. We remain on guard until the situation stabilizes, and all victims are taken care of (Pharmacist 6, Male)."*

*Subtheme 2(a): Personal response*

Personal responses in managing TRDs are indispensable to the overall success of disaster management. Effective and coordinated actions taken by pharmacists save lives, reduce harm, and strengthen resilience in the face of adversity.

However, pharmacists also face security risks and are susceptible to developing post-traumatic anxiety, stress, and depression. These psychological impacts can adversely affect their physical, mental, and emotional well-being, ultimately reducing their efficiency in disaster response. Therefore, it is equally important to consider the psychological condition of pharmacists in such situations to ensure their well-being and maintain the effectiveness of their contributions.

> *"The moment victims arrive at the TC, crowds of attendees pour in and overwhelm the place, making everything more chaotic. The atmosphere feels unsafe, beyond words. I try hard to keep myself steady, but those moments are deeply scarring. For several days afterward, I couldn't eat properly, and I struggled with constant sleep problem." (Pharmacist 3, Male)*

**Theme 3: Preparing for terrorism-related disaster events.** Ideally, healthcare systems must be equipped to handle any type of emergency, including TRDs. In such scenarios, preparedness becomes crucial for minimizing loss of life, reducing psychological trauma, improving coordination, and supporting long-term recovery. Moreover, effective preparation allows for a swift and organized response, limits the social and economic fallout, and strengthens community resilience. By being proactive, healthcare providers and systems can ensure they are ready to protect public safety and contribute meaningfully to community healing and recovery in the aftermath of a terrorist event.

In this study, we focused on the preparedness of pharmacists and their familiarity with terrorism-related disaster management. The pharmacists participating in the study expressed significant concerns about their knowledge and familiarity with managing TRDs. Moreover, all respondents acknowledged the lack of training, seminars, or workshops on the subject, and they did not foresee any such initiatives soon. Additionally, the pharmacists in the study indicated that their role in terrorism-related preparedness is limited. Inline, Theme 3 and its subthemes discuss pharmacists' preparedness for TRDs.

*Subtheme 3(a): Knowledge and understanding of terrorism-related disaster management protocol and policies*

Effective disaster management requires a thorough understanding of management protocols and policies to mitigate the risks associated with such events. The pharmacists involved in the research expressed significant uncertainty regarding their knowledge and awareness of TRDM.

> *"I've been working at the TC for almost five years, but I haven't received any formal training in disaster management. There have been no training modules, resources, or written guidelines provided by the administration. So, most of what I know comes from my own experience in handling terrorism-related emergencies." (Pharmacist 6, Male)*

*Subtheme 3(b): The gaps in managing terrorism-related disasters*

In the event of a terrorism-related event, the primary responsibility of a pharmacist is to ensure the availability and continuous delivery of pharmaceutical supplies. While the American Society of Health-System Pharmacists recommends pharmacists be actively involved in selecting appropriate medications and managing their supply [38,39], this involvement was not observed among the respondents in the current study. Although the pharmacists acknowledged the critical importance of ensuring the availability and timely provision of medicines and relevant supplies to save lives, they also expressed significant reservations that hinder their preparedness when responding to TRDs.

> *"At the TC, we don't have enough staff to cope with the sheer number of emergencies, and the space is already too limited. When there's a large-scale terrorism incident, we have no choice but to use other areas of the hospital, and sometimes we even refer patients elsewhere. But no matter how tough it gets, we make every effort to handle the overload because in the end, it's about saving lives." (Pharmacist 9, Male)*

### Theme 3(c): Triage, assessment and coordination

Triage and coordination aid in prioritizing resources. These are of high importance in resource-deprived settings that help in saving lives by guaranteeing an effective response. Triage categorizes patients based on their injuries and classifies victims needing immediate, intermediate, or minimal care. Consequently, well-organized triage systems are imperative for enhancing disaster preparedness and resilience, including TRDs [40].

The study respondents demonstrated a lack of awareness of the term 'triage' and insufficient knowledge of treatment prioritization, which they attributed to limited training in disaster management. Pharmacists identified challenges in coordination and communication. They acknowledged that coordination and communication within the TC functioned satisfactorily but expressed concerns regarding interactions with other hospital departments. They explained that receiving simultaneous instructions from both the TC administration and the hospital administration often generated confusion and communication conflicts.

*"Triage? (nodding). When victims come in, we examine them right away and begin with basic life support, ACLS, or CPR as needed. But there are no standard treatment protocols we follow. To be honest, I don't even know what triage is; I've never come across the term before." (Pharmacist 1, Male)*

The clarification makes it apparent that both individuals and the institution are missing critical disaster training and standardized triage procedures. Pharmacists' reliance on ad hoc clinical judgment reflects the lack of structured preparedness mechanisms and clear role integration within multidisciplinary emergency response systems.

**Theme 4: Barriers towards terrorism-related disaster management.** *Subtheme 4(a): Security issues (internal and external)*

The barriers to effective TRDs management are provided by theme 4 and the related themes. All pharmacists uniformly reported that victims' attendants display threatening and aggressive behaviour, which places the safety of healthcare professionals at high risk in the TC during a terrorist attack. A pharmacist quoted that *"such an aggravated crowd destroys institutional assets and even physically tortures healthcare professionals (Pharmacist 8, Male)."* Therefore, the institutional administration and health authorities must ensure the safety of the institute and the healthcare workers as a priority so they can work efficiently without fear.

Secondly, following a terrorism-related event, particularly a suicide attack, volunteers and victims' relatives often rush to the hospital, creating overcrowded conditions. Such congestion can make the hospital an easy target for a secondary attack. Therefore, controlling the influx of people into the hospital is crucial to prevent further incidents.

*"I'll never forget the attack on the principal of the local law college. When his body was brought here and the lawyers gathered, it turned into a massive assault that killed many people. In moments like that, I find it so hard to focus on my work. There's always the fear of another suicide attack breaking out in the crowd. That fear weighs heavily on us and makes it harder to provide emergency care." (Pharmacist 8, Male)*

### Subtheme 4(b): Curriculum deficiencies

The pharmacist at the TC explained that the pharmacy curriculum in Pakistan does not include disaster management, particularly terrorism-related disaster management, which creates another barrier. Even after the degree is awarded, there is no residency program offered to the pharmacists. Although the TC on its own offers an internship program to pharmacists, this initiative is not enough. Notably, pharmacists have no idea of disaster management when they start pharmacy practice at a healthcare institute.

*"In pharmacy school, we never learned about disaster management, and even in our hospital internships, the topic was never brought up. But given what we face today, it's so important that pharmacists are trained properly in this area." (Pharmacist 10, Female)*

### Subtheme 4(c): Inadequate experiential learning opportunities

Ultimately, the absence of disaster drills and inclusive training emerged as a significant obstacle to effective disaster management. Unfortunately, the TC has not implemented any disaster drills, nor have the pharmacists participated in comprehensive training or workshops aimed at enhancing their disaster management skills, including TRDs. Disaster drills and exercises play a crucial role not only in skill development but also in building confidence, which leads to improved efficiency and effectiveness during emergencies, particularly in terrorism-related incidents.

*"At this hospital, I've never seen a single training session on disaster management. We've never been part of drills or mock exercises to sharpen our response. Day to day, we just do our routine work as usual, with no real preparation in place for disaster situations." (Pharmacist 9, Male)*

Summarizing, pharmacists' preparedness was largely shaped by experiential exposure rather than formal training or policy guidance. While experiential learning enabled adaptive responses, it simultaneously exposed systemic gaps, including the absence of written protocols, standardized triage systems, and clearly defined professional roles. This reliance on experience underscores an institutional dependence on individual resilience in place of structured disaster preparedness.

### Subtheme 4(d): Minimal pharmacist contribution to planning and management activities

The study respondents also identified an additional barrier. Like other areas in the province, pharmacists at the TC are primarily focused on supply chain and logistics despite their multifaceted roles. As a result, many of their other potential contributions may be overlooked, leading to an underdeveloped pharmacy response to disasters. This barrier could ultimately increase the risk and impact of terrorist activities.

*"When it comes to terrorism-related disaster planning, pharmacists aren't involved. It's not because we lack capability, but because our role still isn't clearly outlined in specialized services such as emergency response frameworks." (Pharmacist 4, Female)*

**Theme 5: Suggestions and recommendations.** Pharmacists of our study unanimously emphasized the need for significant improvements in terrorism-related disaster management (Theme 5). Key recommendations included are as follows:

a) Upgrading and expanding the TC with respect to human resources, physical infrastructure, medical supplies, and logistics,

b) Establishing an independent procurement system for medicines and related items,

c) Developing standard operating procedures for TRDs, and

d) Strengthening and expanding the role of pharmacists in disaster preparedness and response.

Moreover, pharmacists recommended implementing advanced communication and coordination systems, organizing specialized training programs for pharmacists, conducting regular mock drills, and revising the pharmacy curriculum to better address TRDs in the future.

## Discussion

Over the past decades, terrorism has affected nearly every region of the world, with certain nations more frequently confronted with TRDs [41]. In this global context, Pakistan, as a frontline ally in the war against terror, has endured recurrent terrorist attacks. An escalation of TRDs was observed in the aftermath of 9/11, with Pakistan emerging as the third most

affected nation, experiencing a substantial increase of 12% in incidents between 2002 and 2009 [10]. This trend persisted until 2011, after which a modest decline was noted. However, the country consistently remained among the top five most affected countries, ranking fourth in 2016 and fifth in 2019 [42,43]. Regrettably, the Global Terrorism Index 2025 reported Pakistan as the second-most affected country, with a marked rise in TRDs compared to previous years [7].

Each time a terrorism-related disaster occurs in Pakistan, the entire nation bears the burden. Nevertheless, certain institutions are profoundly affected, with the healthcare system serving as a prime example. We must remember that the role of healthcare in managing TRDs is multifaceted, including emergency medical services to treat mass casualties, inter-agency coordination, psychosocial support, and public health protection. Therefore, in a conflict-driven country like Pakistan, a prepared and resilient healthcare system is required that is ready to mitigate the immediate, short- and long-term consequences of terrorism [44]. An efficient healthcare system not only reduces mortality and complications but also helps manage public anxiety and trauma. Accordingly, this qualitative study was conducted to investigate the preparedness and response strategies employed by pharmacists, recognized as the third pillar of the healthcare system, in addressing TRDs in Quetta, Pakistan.

Several inconsistencies were noted during the data analysis and informal discussions. Although the trauma pharmacists' commitment and frontline involvement during TRDs were understandable and commendable, training gaps, poor role definition, and lack of support undermined their full potential while dealing with TRDs. In a country faced with terrorism, we urge that the respondents' insights and observations warrant critical reflection, as pharmacists are integral in responding to TRDs and their presence is fundamental in emergency settings. This is evident from the study findings, which show that even in the absence of formal disaster training, respondents actively provided critical pharmaceutical services, coordinated medication distribution, and ensured the operational flow of supplies during emergencies. This validates pharmacists' value in disaster contexts, including TRDs, and highlights the need for pharmacists' expertise beyond traditional dispensing and supply roles [16,45,46].

The analysis also revealed that pharmacists' preparedness towards management of TRDs was principally experiential. Their preparedness was driven by past exposures to TRDs rather than by formalized or established protocols. This aligns with the findings of McCourt and colleagues, who, in their systematic literature review, concluded that practicing pharmacists and pharmacy students were least prepared in responding to disasters. Furthermore, the authors highlighted that pharmacists' current involvement and expected roles in disaster management remain unclear. Without a prepared pharmacy workforce and active pharmacy participation in disaster management, critical skill and service gaps may arise, potentially compromising patient care [14]. Similarly, pharmacists of the current study were unfamiliar with the concept of "triage", a fundamental component of disaster care. In summary, the gap in preparedness raises concerns about the consistency and quality of response during high-casualty events and must be addressed seriously by the policy makers and administrators in Pakistan. However, pharmacists' unfamiliarity with triage and fragmented coordination should not be interpreted as individual competency deficits. Instead, these findings reflect systemic exclusions of pharmacists from disaster planning, training, and command hierarchies, an omission that undermines the effectiveness of multidisciplinary responses during high-casualty TRDs.

During the data analysis, the absence of structured training programs, disaster drills, and continuing education on TRDs emerged as a recurrent theme. These findings align with prior studies, which also report limited disaster education in pharmacy programs [47,48]. Disaster management is neither integrated into the curriculum nor reinforced through institutional training. Notably, although the Higher Education Commission (HEC) of Pakistan recently revised the national pharmacy curriculum [49], the term 'disaster' appears only once in the entire document, with no reference to disaster management, whether natural or man-made. Consequently, the curriculum fails to provide students with a fundamental understanding of disasters and their management. Consequently, the curriculum fails to provide students with the fundamental understanding of disasters and their management. This gap extends into professional training, as disaster management is not discussed or practiced during hospital internships. We therefore recommend that disaster management

protocols and procedures be formally integrated into the pharmacy curriculum as a mandatory component. At present, this can be achieved by approving disaster management-related content and incorporating it into the curriculum as a bylaw. This would ensure that students acquire essential knowledge and preparedness to manage disasters during both their academic training and future professional practice.

Within this context, there are evidence-based models that can provide guidance integrating disaster preparedness into pharmacy education in Pakistan. Montana and colleagues developed and evaluated an elective course on the pharmacist's role in disaster management in France. The authors concluded that the course demonstrated the potential to increase the number of pharmacists prepared to respond to disasters. The course was also able to stimulate students' interest in emergency preparedness [47]. Similarly, a video-based education for disaster management provided a safe way to prepare students who might be called upon to work in emergency conditions in Japan [48]. Likewise, Watson et al., carried out a table-top exercise among pharmacists that resulted in improved understanding of disaster management [50]. Drawing from these examples, educationalists and policymakers in Pakistan can plan integrating disaster preparedness as a credit-bearing module within the Pharm-D curriculum with clear competency outcomes. Parallel, adopting blended learning and embedding interprofessional exercises during internships can help in defining roles and improve coordination during terrorism-related mass-casualty incidents.

Among the multiple barriers to effective TRDs, key challenges included security concerns, communication breakdowns, infrastructure limitations, and unclear role delineation. Trauma pharmacists reported instances where institutional chaos and hostile crowds not only compromised patient care but also endangered their personal safety. These findings align with existing literature highlighting the dual physical and psychological vulnerabilities experienced by healthcare providers during TRDs [51]. Despite clear readiness towards TRDs, pharmacists also complained that they were largely excluded from disaster planning and coordination. Their involvement as healthcare professionals is only reactive and confined to logistical support. This contrasts with recommendations by the American Society of Health-System Pharmacists and the International Pharmaceutical Federation, whereby the organizations advocate for pharmacists' inclusion in all disaster phases [12,13]. Based on the current findings and recommendations of international governing bodies, there is an urgent need to redefine the role of pharmacists within national and institutional disaster frameworks. Actively involving pharmacists in triage decisions, coordination meetings, and protocol development could substantially improve system efficiency and disaster preparedness and should therefore be prioritized by policymakers in Pakistan.

## Conclusion

Pharmacists play a pivotal but underrecognized role in responding to TRDs. While their commitment is evident, systemic and educational shortcomings severely limit their effectiveness. By addressing barriers in training, institutional support, and disaster role clarity, healthcare systems can harness the full potential of pharmacists as frontline responders in complex emergencies. Findings of the current study underscore the need for healthcare systems in conflict zones to expand disaster management roles to include pharmacists more comprehensively, improving both immediate responses and long-term recovery efforts.

## Limitations

This study is context-specific to a single TC in Quetta, Pakistan, and may not be generalizable to all healthcare institutions or geographic settings. Four pharmacists declined participation primarily due to work schedule constraints and competing clinical responsibilities. While non-participation may have limited the inclusion of some perspectives, data saturation was achieved by the eighth interview, and two additional interviews were conducted to confirm thematic stability. Nonetheless, the possibility that non-participants may have held differing views cannot be entirely excluded and should be considered when interpreting the findings.

## Implications for practice and policy

To bridge the preparedness gap, institutional and educational reforms are needed. First, pharmacy curricula in Pakistan and similar settings should incorporate disaster and terrorism-related modules, emphasizing emergency protocols, triage, and psychosocial care. Second, hospitals must institutionalize disaster drills and continuous professional development sessions to enhance response competencies. Third, policymakers should formally integrate pharmacists into disaster preparedness and planning teams to maximize their contribution across the care continuum. These interventions are particularly vital in regions with recurrent terrorist threats, where healthcare resilience is paramount. Given the global rise in complex emergencies, a well-prepared pharmacy workforce is not only a policy priority but a public health imperative.

## Supporting information

**S1 File. Coreq.**
(DOCX)

## Author contributions

**Conceptualization:** Fahad Saleem, Mohammad Bashaar.

**Data curation:** Fazal Ur Rehman Khilji, Sajjad Haider, Qaiser Iqbal.

**Formal analysis:** Fahad Saleem, Baharudin Ibrahim, Fatiha Hana Shabaruddin.

**Methodology:** Fahad Saleem, Fazal Ur Rehman Khilji, Fatiha Hana Shabaruddin.

**Project administration:** Mohammad Bashaar.

**Supervision:** Fahad Saleem.

**Writing – original draft:** Fazal Ur Rehman Khilji, Sajjad Haider, Qaiser Iqbal.

**Writing – review & editing:** Fahad Saleem, Baharudin Ibrahim, Fatiha Hana Shabaruddin, Mohammad Bashaar.

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
