## [Decision Letter · Decision Letter 0]

8 Dec 2025

Dear Dr.  Bashaar,

Thank you for submitting your manuscript to PLOS ONE. After careful consideration, we feel that it has merit but does not fully meet PLOS ONE’s publication criteria as it currently stands. Therefore, we invite you to submit a revised version of the manuscript that addresses the points raised during the review process.

We look forward to receiving your revised manuscript.

Kind regards,

Ali Ahmed, PhD

Academic Editor

PLOS ONE

Journal Requirements:

Reviewer's Responses to Questions

**Comments to the Author**

1. Is the manuscript technically sound, and do the data support the conclusions?

Reviewer #1: Partly

Reviewer #2: Yes

Reviewer #3: Yes

2. Has the statistical analysis been performed appropriately and rigorously?

Reviewer #1: I Don't Know

Reviewer #2: Yes

Reviewer #3: N/A

3. Have the authors made all data underlying the findings in their manuscript fully available?

Reviewer #1: Yes

Reviewer #2: Yes

Reviewer #3: Yes

4. Is the manuscript presented in an intelligible fashion and written in standard English?

Reviewer #1: No

Reviewer #2: Yes

Reviewer #3: Yes

Reviewer #1: The manuscript provides rich qualitative insights, but several themes—especially those related to triage and coordination—would benefit from clearer linkage between participants’ quotes and the broader systemic gaps, helping to strengthen the analytical depth.

Although the Introduction effectively establishes the significance of terrorism-related disasters in Pakistan, it could be strengthened by more explicitly demonstrating how pharmacists' roles differ from or complement other healthcare professionals in these settings.

The Methods section is generally strong, but a more detailed justification for the universal sampling approach—and discussion of how non-participation (4/14 pharmacists) may have influenced findings—would improve methodological transparency.

The Discussion clearly outlines training and curriculum gaps; however, further comparison with models from other countries that have successfully integrated disaster preparedness into pharmacy education could provide more actionable, evidence-based recommendations.

Reviewer #2: As I mentioned in my attached comments, such studies relate to the preparedness of pharmacists in response to the needs of war or terrorism-hit regions. A pharmacist is a healthcare provider and should be ready to tackle all emergencies, regardless of what they are. So I think the study is a kind of awareness and alerting one. It is well designed and presented. The conclusion effectively summarises the key findings of the study.

Reviewer #3: The manuscript “Beyond the Counter: Pharmacists Preparedness and Response Strategies in Terrorism-Related Emergencies in Quetta, Pakistan” presents a qualitative study exploring pharmacists experiences and preparedness during terrorism-related emergencies. Overall the study addresses an important topic. Please see the attached file for detailed reviewer comments.

**Do you want your identity to be public for this peer review?** For information about this choice, including consent withdrawal, please see our Privacy Policy

Reviewer #1: **Yes:** Nasser M Alorfi

Reviewer #2: **Yes:** Prof. Dr. Iyad Naeem Muhammad

Reviewer #3: No

---

## [Author Response · Author response to Decision Letter 1]

5 Jan 2026

We sincerely thank you and the reviewers for the time and effort invested in evaluating our original submission and for the constructive and insightful comments provided. We have carefully considered all reviewer and editorial comments and have revised the manuscript accordingly. All comments have been fully addressed, and the changes made are detailed point-by-point in the accompanying response to reviewers document. Revisions in the manuscript have been clearly highlighted to facilitate review.

---

## [Editor Report · Decision Letter 1]

21 Jan 2026

Beyond the Counter: Pharmacists’ Preparedness and Response Strategies in Terrorism-Related Emergencies in Quetta, Pakistan

PONE-D-25-55815R1

Dear Bashaar,

We’re pleased to inform you that your manuscript has been judged scientifically suitable for publication and will be formally accepted for publication once it meets all outstanding technical requirements.

Kind regards,

Ali Ahmed, PhD

Academic Editor

PLOS One
---

## [Editor Report · Acceptance letter]

PONE-D-25-55815R1

PLOS One

Dear Dr. Bashaar,

I'm pleased to inform you that your manuscript has been deemed suitable for publication in PLOS One. Congratulations! Your manuscript is now being handed over to our production team.

Kind regards,

on behalf of

Dr. Ali Ahmed

Academic Editor

PLOS One